# Feasibility of Predicting Static Dielectric Constants of Polymer Materials: A Density Functional Theory Method

**DOI:** 10.3390/polym13020284

**Published:** 2021-01-17

**Authors:** Zheng Tang, Chaofan Chang, Feng Bao, Lei Tian, Huichao Liu, Mingliang Wang, Caizhen Zhu, Jian Xu

**Affiliations:** Institute of Low-Dimensional Materials Genome Initiative, College of Chemistry and Environmental Engineering, Shenzhen University, Shenzhen 518060, Guangdong, China; tangzheng1104@163.com (Z.T.); cfchang1104@163.com (C.C.); bfisvip@163.com (F.B.); leitian@szu.edu.cn (L.T.); huichaoliu@szu.edu.cn (H.L.); jxu@iccas.ac.cn (J.X.)

**Keywords:** dielectric constant, polymer materials, density function theory

## Abstract

The rapid development of electronic devices with high integration levels, a light weight, and a multifunctional performance has fostered the design of novel polymer materials with low dielectric constants, which is crucial for the electronic packaging and encapsulation of these electronic components. Theoretical studies are more efficient and cost-effective for screening potential polymer materials with low dielectric constants than experimental investigations. In this study, we used a molecular density functional theory (DFT) approach combined with the B3LYP functional at the 6-31+G(d, p) basis set to validate the feasibility of predicting static dielectric constants of the polymer materials. First, we assessed the influence of the basis sets on the polarizability. Furthermore, the changes of polarizability, polarizability per monomer unit, and differences in polarizability between the consecutive polymer chains as a function of the number of monomers were summarized and discussed. We outlined a similar behavior for the volume of the polymers as well. Finally, we simulated dielectric constants of three typical polymer materials, polyethylene (PE), polytetrafluoroethylene (PTFE), and polystyrene (PS), by combining with the Clausius–Mossotti equation. The simulated results showed excellent agreement with experimental data from the literature, suggesting that this theoretical DFT method has great potential for the molecular design and development of novel polymer materials with low dielectric constants.

## 1. Introduction

Nowadays, modern electronic devices and products are developing in the direction of lightness, thinness, high performance, and multifunctionality, following the demands of the growing electronic industry. Since the characteristic size of an electronic component gradually decreased, i.e., the integration level increased, the resistance–capacitance (RC) delay became larger, yielding a series of problems, such as information transmission delay, increased noise interference, and increased power dissipation, which significantly limited the final performance [1,2]. The design and development of novel materials with low dielectric constants to replace traditional dielectric media, such as silicon dioxide, is an effective method to overcome the aforementioned issues. Compared with conventional dielectric materials, polymer materials attract considerable attention as dielectric materials because of their easy processing, flexibility, tailorable properties for specific uses, and excellent chemical resistance [2].

Generally, weak polar or nonpolar polymer materials are desirable for achieving a low dielectric constant. For the weak polar or nonpolar polymer materials, the relationship between the dielectric constant and the external electric field can be quantitatively expressed by the Clausius–Mossotti equation [1,2]. This equation shows that the dielectric constant depends on the polarizability and volume of the polymer material, implying that the known values of the polarizability and volume of the polymer material can be used to predict its dielectric constant. 

Ab initio methods are most widely used to simulate the polarizability and volume of polymer materials. At a microscopic level, specific properties of the polymer materials are usually modeled by increasing the polymer chain length until the investigated values reach saturation. However, relatively large polymer chains need to be adopted to achieve the saturation level, which increases the computational cost. Therefore, a cost-effective method is needed. Compared with Hartree–Fock methods and post-Hartree–Fock methods, density functional theory (DFT) methods have an acceptable accuracy and moderate computational cost. They can be applied to calculate the dielectric properties of relatively large polymer chains. A literature survey indicates that molecular mechanics methods were commonly used to study the dielectric behavior of polymer materials [3,4]. In contrast, the molecular approach using polymer chains for calculations was rarely reported. Ruuska et al. [5] used a pure DFT method to calculate the dielectric constant of polypropylene (PP). They applied the Perdew–Wang exchange and correlation functional (PW91) [6] and the 6-311++G(d, p) basis set together with the Clausius–Mossotti equation [1]. Although the calculated value (2.52) is very close to the experimental one (2.2–2.3) [5], it still indicates a slight overestimate. This can be ascribed to the overestimates in polarizability, which are inherent in the calculations based on the pure DFT methods combined with the local density approximation (LDA) and generalized gradient approximation (GGA) for the exchange–correlation functional [7,8,9]. Hybrid functionals, where the nonlocal HF exchange is partly mixed in the DFT calculation, can effectively avoid such overestimates and provide a more accurate dielectric constant of polymer materials [10,11,12].

This study’s primary purpose is to explore the feasibility of a molecular DFT approach combined with hybrid functionals in order to reliably predict the dielectric constant of polymer materials. We investigated three typical polymer materials: polyethylene (PE), polytetrafluoroethylene (PTFE), and polystyrene (PS). The DFT method, in connection with the hybrid functional B3LYP [13] referring to Becke’s three-parameter functional [11,12] combined with the Lee–Yang–Parr correlation functional, was employed [10]. First, we calculated the polarizability and volume of the polymer chains with different numbers of monomers for PE, PTFE, and PS polymer materials. Second, the influence of the basis set on the polarizability was presented. The changes of the polarizabilities, polarizabilities per monomer unit, and differences in polarizability between the consecutive polymer chains as a function of the number of monomers were summarized and discussed. Similarly to these assessments, we outlined the changes in the volume of the polymer materials as well. Finally, the dielectric constants of the polymer materials were calculated and compared with the experimental values from the literature. In conclusion, we discussed the feasibility of the molecular DFT approach combined with the B3LYP functional at the 6-31+G (d, p) basis set for reliably predicting the dielectric constants of the chosen polymer materials.

## 2. Computational Methods

The Clausius–Mossotti equation [1] can be applied for modeling the dielectric constants of polymer materials if we assume that the polymer materials are composed of identical, nonpolar polymer chains and that chain–chain interactions are negligible. It provides an approximate analytical relation between the dielectric constant and the polarizability of the model molecules. The derivation of the Clausius–Mossotti equation is outlined in the Appendix A. Generally, the Clausius–Mossotti equation can be written as follows:(1)$$\frac{\varepsilon_{r}-1}{\varepsilon_{r}+2}\frac{M_{w}}{\rho }=\frac{N_{A}\alpha }{3\varepsilon_{0}}$$
where *ε_r_* is the dielectric constant, *M_w_* is the molar mass of the dielectric medium, and *ρ* is its density. *N_A_* is the Avogadro’s constant, α is the polarizability, and *ε*_0_ is the permittivity of free space. Since the dielectric constant *ε_r_* does not have units, the right term *N_A_**α/*3*ε*_0_ of Equation (1) has units of volume. It is worth noting that Equation (1) is only applicable to the international system (SI) of units. However, for the centimeter–gram–second system (CGS) of units, the Clausius–Mossotti equation must be rewritten as follows:(2)$$\frac{\varepsilon_{r}-1}{\varepsilon_{r}+2}=\frac{4\pi N_{A}\alpha^{\prime }}{3V_{m}}$$
where *V_m_* is the molar volume of the dielectric medium, $\alpha^{\prime }$ is the molecular polarizability volume defined in terms of the conventional polarizability α as $\alpha^{\prime }\equiv \alpha /4\pi \varepsilon_{0}$ [1,14]. From Equation (1) or (2), we can see that the dielectric constants of the polymer materials can be predicted if we can accurately calculate the polarizability and volume of the polymer material.

In this article, we performed all calculations with the Gaussian16 software package [15] to accurately simulate the polarizability and volume of the polymer materials. The DFT method, in connection with the B3LYP [13] referring to Becke’s three-parameter functional [11,12] combined with the Lee–Yang–Parr correlation functional, was employed [10]. We used a series of split-valence basis sets. The employed split-valence basis sets were 6-31G**, 6-31+G**, and 6-31++G**, where the ** denoted the (d, p) polarization functions, while the ++ stood for the diffuse functions for heavy atoms and hydrogens [16]. All the properties, including the polarizability and volume, were calculated at the same theoretical level at which their geometry was optimized.

The elements of the static linear polarizability tensor $\alpha^{ij}\left(0\right)\equiv \alpha^{ij}\left(-\omega ,\omega \right){|}_{\omega =0}$ were calculated using a pseudo-energy derivative approach proposed by Rice and Handy [17]:(3)$$\alpha^{ij}\left(-\omega ,\omega \right){|}_{\omega =0}=-{\frac{\partial^{2}W}{\partial E_{0i}\partial E_{\omega j}}|}_{\omega \to 0}$$
where *W* is the pseudo-energy of the molecule. $E_{0i}$ and $E_{\omega j}$ are the static and dynamic components of the external time-dependent electric field, respectively. 

The isotropic or mean polarizability is defined as an average of the diagonal elements of the polarizability tensor, i.e., $\alpha_{iso}=1/3\left(\alpha_{xx}+\alpha_{yy}+\alpha_{zz}\right)$. In this article, we set the polymer chains along the *x*-axis direction. Thus, the longitudinal polarizability component along the chain is $\alpha_{//}=\alpha_{xx}$. The mean value of the two transverse components of the polarizability tensor was also used, i.e., $\alpha_{⊥}=1/2\left(\alpha_{yy}+\alpha_{zz}\right)$. Moreover, we presented both the polarizability per monomer unit ($\alpha^{\prime }\left(N\right)/N$) and the differences in polarizability for consecutive polymer chains ($\alpha^{\prime }\left(N\right)-\alpha^{\prime }\left(N-1\right)$) to better reveal the change in the polarizability change with the polymer chains’ length.

The volume of the polymer chains was calculated using the polarizable continuum model (PCM) [18], which is a widely used implicit solvation model. The cavity volume covered by the solvent accessible surfaces (SAS) [19,20] was defined as the polymer chain volume.

## 3. Results and Discussion

### 3.1. Polarizability

Generally, a relatively large basis set needs to be selected to achieve the convergence in polarizability. However, the size of the basis set is always proportional to the computational cost, so that the balance between the computational cost and precision must be considered during the selection of the basis set. Three basis sets, including 6-31G(d, p), 6-31+G(d, p), and 6-31++G(d, p), were employed during the simulations to better indicate the influence of the basis sets on the polarizability of the selected polymers. Figure 1 shows the simulated isotropic polarizability as a function of the number of monomers for different polymer materials with different basis sets. We can see that the polarizability gradually converges when increasing the basis sets for all polymer materials. Furthermore, there is only a slight (Figure 1a) or no (Figure 1b,c) improvement in the precision of the polarizability gained by increasing the basis sets from 6-31+G(d, p) to 6-31++G(d, p). Therefore, since the basis set 6-31++G(d, p) was more time-consuming than the basis set 6-31+G(d, p), we chose the 6-31+G(d, p) basis set to further simulate the polarizability.

In addition, Figure 2 shows both the longitudinal ($\alpha_{//}^{\prime }$) and mean transverse ($\alpha_{⊥}^{\prime }$) components of the polarizability and its isotropic value to better understand the influence of the polymer chain’s length on the polarizability. From Figure 2a–c, three features can be defined. First, all the polarizability components seem to depend linearly on the number of monomers, i.e., the length of the polymer chain, which is the typical behavior of saturated polymers [5,21]. Second, the slopes of the isotropic, longitudinal, and mean transverse polarizabilities are quite different. The longitudinal component slope is the largest, followed by the slope of the isotropic polarizability, while the longitudinal component exhibits the smallest slope. The fact that the longitudinal polarizability increases significantly faster than the mean transverse component will certainly cause the polarizability anisotropy, affecting the dielectric constant of the polymer materials. These influences will be discussed in detail in later sections. Finally, as PS has a relatively large monomer, there seem to be apparent changes in the slopes of both the longitudinal and the mean transverse polarizability components. As shown in Figure 2c, small deviations from the linearity occur. This might be attributed to the coupling effect of the longitudinal and the mean transverse components, while when the longitudinal component increases, the mean transverse component decreases, and vice versa. However, the isotropic polarizability slope is almost unchanged (see the gray square in Figure 2c).

Furthermore, in Figure 3 we plotted the isotropic polarizability per monomer unit ($\alpha^{\prime }\left(N\right)/N$), and the longitudinal ($\alpha_{//}^{\prime }/N$) and mean transverse ($\alpha_{⊥}^{\prime }/N$) components as a function of the chain length for all the polymer materials in order to get a better insight into the slope changes. Herein, three distinct features can be observed. First, for PE and PTFE, the longitudinal components increase regularly with the molecular chain length, while the growth gradually saturates later. Conversely, the mean transverse components monotonously decrease with the chain length until they slowly saturate toward constant values. However, PS exhibits quite a different behavior, as shown in Figure 3c. There are no monotonous increasing and decreasing trends of the longitudinal and transverse components. Second, the coupling effects mentioned above between the longitudinal and transverse components are significantly amplified. When the longitudinal components increase, the mean transverse components decrease simultaneously, and vice versa. Indeed, there is an exception to this behavior, as shown in Figure 3b. When the number of monomers increases from one to two, both the longitudinal and transverse components decrease, which might be attributed to the finite-size quantum mechanical effects of small molecules. Finally, it is remarkable that the isotropic polarizabilities per monomer unit, i.e., $\alpha_{iso}^{\prime }/N=\left[\left(1/3\right)\alpha_{//}^{\prime }+\left(2/3\right)\alpha_{⊥}^{\prime }\right]/N$, remain essentially constant as a function of the molecular chain lengths, although the longitudinal and transverse components of the polarizability are coupled.

Figure 4 shows the differences in polarizability components for the consecutive polymer chains differing by one monomer unit in length, i.e., $\alpha^{\prime }\left(N\right)-\alpha^{\prime }\left(N-1\right)$. On the one hand, in Figure 4a,b we can observe that the longitudinal and mean transverse components tend to saturate gradually. At the same time, the isotropic polarizability remains almost unchanged for PE and PTFE. Compared with the polarizability components per monomer unit, the polarizability components between the consecutive polymer chains saturate more quickly. On the other hand, when comparing Figure 3c with Figure 4c, we can see that the coupling effects between the longitudinal and transverse components are further significantly amplified. Interestingly, there are two distinct crossovers for the longitudinal and the mean transverse polarizability components with three and four monomers. Finally, similar to Figure 3, the longitudinal and the mean transverse polarizability components are highly coupled, yielding the constant isotropic polarizability values of $\alpha^{\prime }\left(N\right)-\alpha^{\prime }\left(N-1\right)$ in Figure 4.

### 3.2. Volume

Similar to the polarizability simulations, we employed three basis sets, 6-31G(d, p), 6-31+G(d, p), and 6-31++G(d, p), during the simulations to better reveal the influence of the basis sets on the volume. The simulated volumes for three typical polymer materials at different basis sets are shown in Table 1. We can see that the influence of the basis sets on the volume is negligible. Therefore, in this article’s later section, to ensure that the volumes were calculated at the same theoretical level of use for the simulation of their polarizability, we used the volumes simulated with a basis set 6-31+G(d, p) to calculate the dielectric constants of the polymer materials.

We show the volume per monomer unit V(N)/N and the volume differences between the consecutive polymer chains V(N)−V(N−1) as a function of the number of monomers N in Figure 5a–c, respectively, to better indicate the influence of the length of the polymer chain on the volume, the volume V(N). From Figure 5b, we can see that the volume per monomer unit decreases monotonously with the molecular chain length and that the effect gradually saturates. Furthermore, the volume differences between the consecutive polymer chains are little (PS) or remain unchanged (PE and PTFE), as shown in Figure 5c.

### 3.3. Dielectric Constant

As we have outlined above, if we can accurately calculate the polarizability and volume of the polymer material, then we can predict the dielectric constants of the materials. In Section 3.2 and Section 3.3, the effects of the basis sets and chain lengths on the volume and polarizability are discussed in detail. Herein, we can calculate the dielectric constants of the materials with Equation (1) or (2). The simulated dielectric constants with different chain lengths for the three polymer materials are shown in Table 2. During the simulation, all the calculations, including the geometry optimizations, polarizability calculations, and volume calculations, are based on the B3LYP/6-31G+(d, p) method. Table 2 shows that the dielectric constants increase regularly with the molecular chain length, while the growth gradually saturates. Furthermore, as shown in the fifth and sixth columns of Table 2, the simulated dielectric constants are very close to the experimental value from the literature.

### 3.4. Double Layers of Polymer Chains

Since polymer materials typically consist of multiple stacked polymer chains, the influence of the multilayer structure on the dielectric constant needs to be considered. In this article, the dielectric constants of the double-layer-stacked PE and PTFE materials were calculated. During the simulation, a geometrically-optimized polymer chain composed of six monomers and the PACKMOL software package [28] were first adopted to model the double-layer-stacked structure. Two polymer chains with interlayer distances of 2.0 and 2.5 Å were modeled, respectively. Second, the geometrical optimizations of these modeled structures were applied. Third, the polarizability, volume, and dielectric constant of these modeled structures were calculated. Table 3 outlines the simulated dielectric constants. Compared with the dielectric constants of single polymer chains, we can see that the dielectric constants of the double-layer-stacked structures are very close to those of the single polymer chains, indicating that the influence of the multilayer structure on the dielectric constant is negligible. Therefore, under the actual conditions, modeling only the single polymer chains is accurate enough to predict the dielectric constant of the polymer materials.

## 4. Conclusions

In this study, we investigated the feasibility of a molecular DFT approach combined with the B3LYP functional to predict the dielectric constant of polymer materials. The simulation results of the three typical polymer materials, PE, PTFE, and PS, show four main findings. First, the 6-31+G(d, p) basis set is accurate for calculating the polarizability and volume of the polymer chains. Second, the isotropic polarizability and volume gradually saturate with the length of the polymer chains. Third, the values of the simulated static dielectric constants derived by combining the simulated polarizabilities and volumes with the Clausius–Mossotti equation show an excellent agreement with the experimental data from the literature. Fourth, the influence of the multilayer structure on the dielectric constant is negligible. These findings suggest that the molecular DFT approach combined with the B3LYP functional at the 6-31+G(d, p) basis set can provide a reliable prediction of the dielectric constants of polymer materials. Besides this, all the calculations can be implemented with conventional supercomputer workstations, and the typical total elapsed time for all the calculations for an individual polymer material only takes a few hours (see Appendix A). Therefore, the method outlined in this study has a feasible computational cost, and it has the potential for high-throughput computing.

Besides this, for the case of PTFE, although the calculated dielectric constant (1.80) is very close to the experimental one (2.0), a slight underestimate still exists. The precise mechanism of this underestimate still remains to be answered. There are two possible factors that might cause this deviation. First of all, it might be attributed to the strong electronegativity of the fluorine atoms. Second, an inappropriate hybrid functional has been chosen. Therefore, to improve the precision of the method, the influences of the electronegativity of atoms and the hybrid functionals on the theoretical dielectric constants need to be further investigated.

In summary, the method presented in this study has the advantages of being simple and feasible, and of having a low computational cost, and it will be very helpful for the molecular design and development of novel polymer materials with low dielectric constants. For the researcher who works on the experimental synthesis, a theoretical simulation of the dielectric constant by this method before the experiment is strongly recommended. It will significantly improve your experimental efficiency and save you experimental costs.

## Figures and Tables

**Figure 1 polymers-13-00284-f001:**
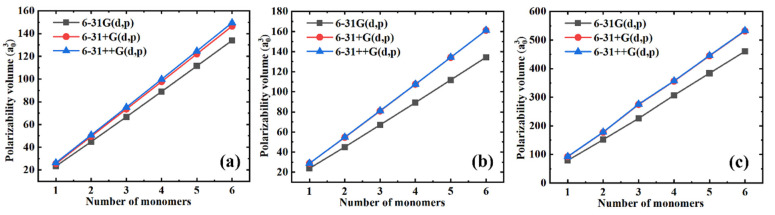
The simulated isotropic polarizability as a function of the number of monomers for different polymer materials with different basis sets. (**a**) PE; (**b**) PTFE; and (**c**) PS.

**Figure 2 polymers-13-00284-f002:**
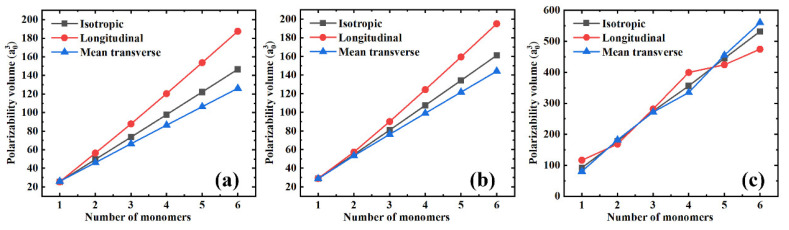
The isotropic, longitudinal, and mean transverse polarizability components as a function of the number of monomers for different polymer materials. (**a**) PE; (**b**) PTFE; and (**c**) PS.

**Figure 3 polymers-13-00284-f003:**
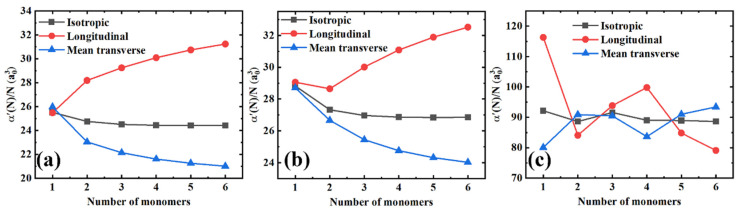
The isotropic, longitudinal, and mean transverse polarizabilities per monomer unit as a function of the number of monomers for different polymer materials. (**a**) PE; (**b**) PTFE; and (**c**) PS.

**Figure 4 polymers-13-00284-f004:**
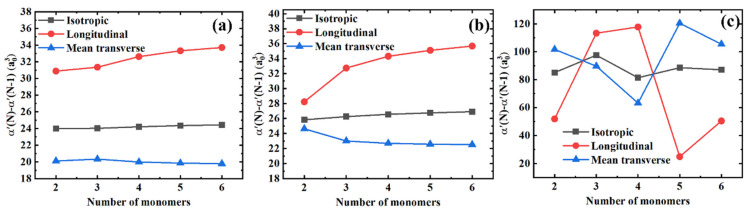
Differences in the isotropic, longitudinal, and mean transverse polarizability components between the consecutive polymer chains for different polymer materials. (**a**) PE; (**b**) PTFE; and (**c**) PS.

**Figure 5 polymers-13-00284-f005:**
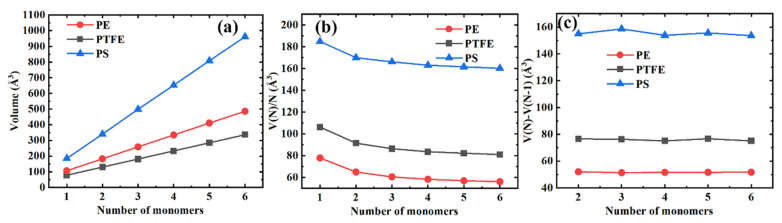
Different volume representations for three typical polymer materials. (**a**) The volume as a function of the number of monomers; (**b**) The volume per monomer unit as a function of the number of monomers; (**c**) Differences in the volume between the consecutive polymer chains.

**Table 1 polymers-13-00284-t001:** The simulated volumes for three typical polymer materials at different basis sets. Here, the volume is given in Å^3^.

Name	Number of Monomers	6-31G(d, p)	6-31+G(d, p)	6-31++G(d, p)
PE	1	77.769	77.815	77.815
	2	129.719	129.833	129.833
	3	181.026	181.191	181.191
	4	232.604	232.793	232.793
	5	284.233	284.463	284.463
	6	335.866	336.146	336.146
PTFE	1	105.953	106.26	106.262
	2	181.521	182.819	183.571
	3	257.166	259.082	258.579
	4	331.783	334.154	334.155
	5	407.344	410.814	410.658
	6	482.5	485.854	485.947
PS	1	184.79	184.79	184.965
	2	339.582	339.582	339.973
	3	495.189	495.189	498.268
	4	650.496	650.496	652.596
	5	805.949	805.949	808.011
	6	960.777	960.777	961.946

**Table 2 polymers-13-00284-t002:** The simulated dielectric constant for molecules with different chain lengths determined using the B3LYP/6-31+G(d, p) method. Here, the units of polarizability and volume are a_0_^3^ and Å^3^, respectively.

Name	Molecule	Polarizability	Volume	Dielectric Constant	Experimental Value
PE	C_2_H_6_	25.51	77.815	1.93	2.2–2.3 [22,23]
	C_4_H_10_	49.51	129.833	2.04	
	C_6_H_14_	73.53	181.191	2.10	
	C_8_H_18_	97.73	232.793	2.13	
	C_10_H_22_	122.08	284.463	2.15	
	C_12_H_26_	146.51	336.146	2.16	
PTFE	C_2_H_2_F_4_	28.82	106.26	1.69	2.0 [24,25]
	C_4_H_2_F_8_	54.65	182.819	1.74	
	C_6_H_2_F_12_	80.9	259.082	1.76	
	C_8_H_2_F_16_	107.46	334.154	1.78	
	C_10_H_2_F_20_	134.21	410.814	1.79	
	C_12_H_2_F_24_	161.12	485.854	1.80	
PS	C_8_H_10_	92.78	184.965	2.48	2.4–2.6 [26,27]
	C_16_H_18_	177.91	339.973	2.51	
	C_24_H_26_	275.64	498.268	2.62	
	C_32_H_34_	356.99	652.596	2.58	
	C_40_H_42_	445.78	808.011	2.59	
	C_48_H_50_	533.14	961.946	2.60	

**Table 3 polymers-13-00284-t003:** The simulated dielectric constants of the single- and double-layer polymer chains for PE and PTFE. Here, the units of interlayer distance, polarizability, and volume are Å, a_0_^3^, and Å^3^, respectively.

Name	Interlayer Distances	Polarizability	Volume	Dielectric Constant	Experimental Value
PE	single layer	146.51	336.146	2.16	2.2–2.3 [22,23]
	2.0	293.54	661.721	2.16	
	2.5	293.68	672.55	2.14	
PTFE	single layer	161.12	485.854	1.80	2.0 [24,25]
	2.0	321.93	969.114	1.79	
	2.5	321.93	969.12	1.79	

## Data Availability

The data presented in this study are available on request from the corresponding author.

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
