# Peer review of "Feasibility of Predicting Static Dielectric Constants of Polymer Materials: A Density Functional Theory Method"

_polymers, 2021, doi:10.3390/polym13020284_

Round 1

Reviewer 1 Report

The Authors have approach to the a theoretical molecular density functional theory approach was used to validate the feasibility of predicting the static dielectric constants of polymer materials. First of all, the influence of the basis sets on the polarizabilities is presented. Besides, the regularities of the polarizabilities, polarizabilities per monomer unit,  and polarizabilities differences between the consecutive polymer chains as the function of the number of monomers are summarized and discussed. The manuscript deserves to publish in Polymers after a major correction. I would like to suggest introducing changes before publishing in Polymers.

The authors should revise in the manuscript as the following points:

  1. Why the Authors used B3LYP functional? Whether another method was taken into account? Are there any links to the experimental data?
  2. For any computational work including topological analysis the authors must supply in Supplementary Materials: Cartesian coordinates of all stationery points determined.
  3. Conclusions should be supplemented with data on future research as well as experimental work.

Reviewer 2 Report

In this manuscript, the authors use hybrid DFT to calculate the dielectric constant of 3 common polymers. They use 3 different basis sets, together with the B3LYP functional. They study the dependence of the results on the basis sets and the number of monomer chains in each polymer. Their results are interesting and can be useful to other researchers interested in calculating polymer dielectric properties using DFT.

I recommend the acceptance of the manuscript AFTER the following issues have been fixed:

1- The authors claim that the second basis set gives accurate results, with reasonable computational cost, using Gaussian. The authors must provide details on the computational setup (processor type, number of cores used in a typical calculation, and how long a calculation typically takes)

2- The equation they use in calculating the dielectric constant using the polarizability is called the Clausius-Mossotti equation. "Mossotti" is misspelled in the abstract and conclusion.

3- The beginning of the "Computational methods" section includes a derivation of the Clausius-Mossotti equation (equations 1 thru 7). This must be removed, as it is text-book information. 

4- Figure 2c has the wrong key. 

5- A revision of the English language is a MUST. For example, the word "chapter" appears multiples times in the manuscript. In addition, some phrases are repeated within the same sentence.

Round 2

Reviewer 1 Report

The authors correctly answered to the questions and the manuscript can be published in the present form.